# Actual Nutrition and Dietary Supplementation in Lithuanian Elite Athletes

**DOI:** 10.3390/medicina56050247

**Published:** 2020-05-20

**Authors:** Marius Baranauskas, Valerija Jablonskienė, Jonas Algis Abaravičius, Rimantas Stukas

**Affiliations:** 1Department of Physiology, Biochemistry, Microbiology and Laboratory Medicine, Institute of Biomedical Sciences, Faculty of Medicine, Vilnius University, 01513 Vilnius, Lithuania; valerija.jablonskiene@mf.vu.lt (V.J.); algis.abaravicius@mf.vu.lt (J.A.A.); 2Department of Public Health, Institute of Health Sciences of the Faculty of Medicine, Vilnius University, 01513 Vilnius, Lithuania; rimantas.stukas@mf.vu.lt

**Keywords:** elite athletes, actual nutrition, diet, dietary supplements, body composition

## Abstract

*Background and objectives:* Health is partly determined by the state of one’s nutrition; it stimulates the body’s functional and metabolic adaptations to physical strain and helps one prevent sports injuries and get in shape in terms of body composition. This study aims to investigate the actual nutrition and dietary supplements taken by elite Lithuanian athletes and to identify the relationship between the dietary intake, dietary supplementation and body composition of elite athletes. *Materials and Methods:* The research subjects were 76.7% of Lithuanian elite athletes (N = 247). The actual diet was investigated using the 24 h recall dietary survey method. Dietary supplementation was studied applying the questionnaire method. Measurements of body composition were performed using the BIA (bioelectrical impedance analysis) tetra-polar electrodes and measuring resistivity with 8–12 tangent electrodes at different frequencies of signal: 5, 50 and 250 kHz. *Results:* Results indicate that among the athletes, 62% use too few carbohydrates and 77% use too much fat. Although the 3.5% increase in lean body mass (95% CI: −0.107, 7.070) helps gain an increased protein intake with food (*p* = 0.057), 38% of athletes consume too little protein with food. The athletes mostly use carbohydrates (86%), vitamins (81%), protein supplements (70%), and multivitamins (62%). We did not determine the impact (*p* > 0.05) of individual or complex supplement use on the lean body mass (%) or fat mass (%) values of athletes. *Conclusions:* Athletes consume insufficient carbohydrates, vitamin D, calcium, polyunsaturated fatty acids, omega-3 and omega-6 fatty acids and too much fat, saturated fatty acids, cholesterol, and they use proteins irrationally. Sport nutritionists should also focus on the risk of malnutrition for female athletes. Nutritional supplements partially offset macronutrient and micronutrient deficiency. Nevertheless, the effect of food supplements on the body composition of athletes is too small compared to the normal diet. Athletes ought to prioritize the formation of eating habits and only then use supplements.

## 1. Introduction

Health is partly determined by the state of one’s nutrition; it stimulates the body’s functional and metabolic adaptations to physical strain and helps one prevent sports injuries and get in shape in terms of body composition [1,2,3,4]. An athlete’s diet should be coordinated with the training process, because then is it possible to improve physical working capacity during the precompetition period and achieve better results in sports competitions [5]. Otherwise, a diet insufficient in carbohydrates, for example, slows the endurance of athletes working to improve their adaptation to physical strain. Incomplete restoration of glycogen storage in the muscles and liver between workouts requires more effort on behalf of the central nervous system to overcome high-intensity physical loads. As a result, the risk of over-training may increase, and the immune system may weaken [6]. A lack of dietary protein means it becomes more difficult to induce muscle hypertrophy and to increase lean body weight [5,7], and while endurance athletes cannot fully ensure positive mitochondrial and sarcoplasmic protein fraction synthesis [8]. Therefore, professional athletes need to get enough energy and nutrients (carbohydrates, protein and fat) with food due to high workloads. Generally, athletes do not have a different recommended daily intake (RDI) for vitamins and minerals, compared to the general population [5].

In addition, food supplements gain a special meaning alongside with the ordinary diet of an athlete. Officially, dietary supplements are commercially available food to supplement a normal diet composed of vitamins, minerals, herbs (botanicals), amino acids, and many other components [9]. Dietary supplement used among athletes in different sports should vary depending on their usage goals. The research shows that the objectives for supplement intake named by athletes are usually the ergogenic effects and recovery after physical activity [5]. Dietary supplements are also often used for health benefits. Athletes use supplements to optimize nutrition by supplementing deficient materials. Supplement users want to improve their immune system and adjust their body composition [10]. However, athletes are known to lack knowledge about dietary supplements: they are not used for their intended purpose and are not compatible with normal eating habits [11,12].

In this context, the research findings about the nutrition and dietary supplementation studies of high-performance athletes are relevant. The data obtained leads to science-based findings and projections that make it possible to optimize the diet and thus more efficiently manage professional athletes at the European and World Championships and the Olympic Games. Specifically, athletes of different sports cultivate rational nutrition combined with a well-organized training process to encourage the maximum adaptation of their bodies to physical precompetition strain and achieve better competition results.

The objective of the study is to characterize the actual nutrition and dietary supplements taken by elite Lithuanian athletes and to identify the relationship between the dietary intake, dietary supplementation and body composition of elite athletes in training processes.

## 2. Material and Methods

### 2.1. Study Population

In order to describe the nutrition, dietary supplementation, and body composition of athletes and determine the effects of different nutrients during exercises developing lean body mass (LBM), in 2016–2017 we selected and analysed 76.7% of Lithuanian elite athletes (N = 247) training for Olympic sports: short- and medium-distance running and hurdles (N = 3), long-distance running (N = 2), weightlifting (N = 6), athletics (jumping, throwing) (N = 5), basketball (N = 42), gymnastics (N = 2), figure skating (N = 2), mountain skiing (N = 3), swimming (N = 41), skiing (N = 14), biathlon (N = 19), kayak/canoe rowing (N = 5), cycling (track, BMX) (N = 6), cycling (road) (N = 21), modern pentathlon (N = 11), boxing (N = 10), judo (N = 11), taekwondo (N = 4), and wrestling (N = 20). Depending on the duration of the physical labour, training, and features of energy production in the body, we classified the athletes in accordance with the sport into three groups: anaerobic (21.9% (N = 54)), mixed aerobic and anaerobic (30.8% (N = 76)), and aerobic (47.4% (N = 117)) [13]. Their athletic experience was 7.9 ± 3.8 years. The average workout was 178.2 ± 63.7 min per day.

The high-performance athletes were tested during the preparatory period before a competition. The scope of the athletes’ workout load fully complied with approved training plans. The assessment of the athletes’ training plans was based on plans used by high-performance athletes in training for the Olympics and officially approved by the Department of Physical Education and Sports and the National Olympic Committee of Lithuania, plans specified in the Rio 2016, Sochi 2014 and PyeongChang 2018 programmes.

### 2.2. Anthropometric Measurements

The height of athletes was measured using a stadiometer (±1 cm) at the Lithuanian Sports Medicine Centre. Measurements of body weight and individual weight components (body mass, LBM (in kilograms and percentage), muscle mass (in kilograms and percentage) and body fat (BF) (in kilograms and percentage) were performed at the Lithuanian Sport Centre using the BIA (bioelectrical impedance analysis) tetra-polar electrodes (13 lot 21 block with certification EN ISO 13488; Jinryang Industrial Complex, Kyungsan City, South Korea) and measuring resistivity with 8–12 tangent electrodes at different frequencies of signal: 1, 5, 50, 250, 550, and 1000 kHz [14,15].

Height and body weight ratios were assessed by calculating the body mass index (BMI). Each athlete’s BMI was determined by calculating it as body weight in kilograms (kg) and dividing by height in squared meters (m^2^). It was assessed according to the standards for describing insufficient body weight (when BMI is <19), normal body weight (when BMI is 19–24), excess weight (when BMI of 24–30), and obesity (when BMI ≥ 30). The muscle and fat mass index (MFMI) of each athlete was determined, which is calculated by dividing the weight of the muscle (in kg) by weight (in kg). The athletes’ MFMI was rated on a scale that describes a very low MFMI (male MFMI < 2 and female < 1.8), a low MFMI (male MFMI corresponding to 2.1–3.39, and female 1.9–2.89), an average MFMI (male MFMI equivalent 3.4–4.69, and female 3–3.99), a large MFMI (with male MFMI corresponding to 4.7–6.0, and female 4–5) and a very high MFMI (with men MFMI > 6 and women > 5) [13].

### 2.3. Dietary Intake

The actual diet of Lithuanian high-performance athletes was estimated using a 24 h recall dietary survey method [16,17,18]. Data on the athletes’ actual diet was collected for the previous day. The survey was performed by a trained interviewer using the direct interview method at the Lithuanian Sports Centre. The interviewer recorded data on food and meals consumed by each athlete. In the course of the food recall, the portion sizes in the special Atlas of Foodstuffs and Dishes were used [19]. The atlas displays different portions of products and meals assessed in grams, making it possible to record the amounts of all food products and meals consumed.

The average daily food intake of athletes was evaluated, and the chemical composition and energy value of the food rations were determined using chemical composition tables [20]. Carbohydrate, protein, and fat intake were assessed according recommendations in academic literature [5,6,21]. The recommended carbohydrate content for athletes having moderate or high intensity daily training of 1–3 h must amount to 6–10 g/kg of body weight [5,6], and protein content must amount to 1.4–2.0 g/kg of body weight [5,21]. The percentage of energy should be between 20% to 35% from fat, <10% from saturated fatty acids (SFA), from 6% to 10% for polyunsaturated fatty acids (PUFA), from 1% to 2% for omega-3 fatty acids (omega-3 FA), and from 5% to 8% for omega-6 fatty acids (omega-6 FA) [5,22]. Athletes (in percentages) were divided into groups by macronutrient intake levels: those consuming less than the RDI, those consuming the recommended amount, and those using more than the RDI.

Athletes’ increased intake of vitamins and/or minerals has little or no effect on physical performance indicators as long as the dietary intake of vitamins and minerals meets the prescribed RDI for a given country. Thus, athletes should consume diets that provide at least the RDI for all micronutrients [5,21,23]. In the case of our study, the same RDI for vitamins and minerals (exclusively, iron and calcium RDI [5,24,25]) of the Lithuanian population [26] were also applied to the analysis of the diets of studied athletes in Lithuania.

### 2.4. Energy Requirements

We estimated the basal metabolic rate (BMR), daily energy expenditure (DEE), and training energy expenditure (TEE) of all subjects. BMR was calculated using the Harris & Benedict formulas [27]. The 24 h period spent on regular and nonregular activities, sedentary activities, and sleeping habits as reported by the American Dietetic Association, Dieticians of Canada, and the American College of Sports Medicine were the physical activity and lifestyle variables considered in this study [23]. Twenty-four-hour records of physical activity were collected on the same day the participants recorded their dietary intake. The studies of Ainsworth et al. supported these measures [28], and data were managed according to the specific activity.

The metabolic equivalent intensity level (MET) was used; it should be understood as the ratio of the metabolic rate while working to a standard resting metabolic rate of 1.0 (4.184 kJ)/kg/h (or 3.5 mL O^2^/kg/min). Participants recorded the 24 h period frequency and duration of the different activities expressed and the rate of energy expenditure for each activity were estimated. The physical activity level (PAL) of all athletes was calculated as the ratio between DEE and BMR as reported by FAO/WHO/UNU [29]. To estimate the total energy requirement (EER), the BMR was then multiplied by the PAL.

### 2.5. Dietary Supplementation

We used a validated (by Baranauskas M. [30]) questionnaire for the dietary supplementation study. The respondents participated in direct interviews. The questionnaire was about athlete socio-demographics (gender, age, place of residence, sport, sporting experience, etc.) and dietary supplementation.

### 2.6. Statistical Analysis

All normally distributed continuous variables are presented as means ± standard deviations (SD), whereas qualitative variables are presented as relative frequencies (in percentage). Normality of variable distribution was tested by the Shapiro–Wilk *W*-test. When normality was confirmed, paired sample *t*-tests were used to assess differences between groups. Associations between categorical variables were tested using contingency tables and the calculation of the χ-squared or Fisher’s exact test. Multiple linear regression analysis was used to test dietary intake predicting LBM% and BF% from dietary intake. Models were adjusted for age, sex, and type of sport. Logarithmic or inverse square transformations were used to improve normality. All reported *p*-values are based on two-sided tests and compared to a significance level of 5%. Statistical analyses were performed using Stata version 12.1 (StataCorp, College Station, TX, USA), SPSS V.25 for Windows (Armonk, NY, USA) and Microsoft Excel (Seattle, WA, USA).

### 2.7. Ethics Statement

All the organizational issues regarding the survey were discussed with the Lithuanian Sports Centre and with the Bioethics Committee prior to the research. The study was conducted in accordance with a permit to carry out biomedical research issued by the Lithuanian Bioethics Committee (No. 158200-11-113-25, from 3 November 2009). Prior to testing, all athletes provided written consent and the study protocols were approved by the Lithuanian Sports Medicine Center Institutional Review Board.

## 3. Results

### 3.1. Nutrition of Athletes

Data describing the actual nutrition of athletes shows that for all the sports both male and female athlete energy intake (EI) (excluding food supplements derived from EI) ranges from 2197.6 kcal to 4244.9 kcal and is less than the EER (range from 3461.7 to 4845.1 kcal) (Table 1).

We determined that 62.4% of athletes do not consume enough carbohydrates. The average carbohydrate intake of male and female athletes ranges within 5.6 ± 1.9 g/kg of body mass per day to 4.8 ± 2.1 g/kg of body mass per day, which amounts to less than the minimum recommended 6.0 g/kg of body mass per day. In addition, unlike male athletes, female athletes often lack carbohydrates in their diet (50.8% vs. 77%, χ^2^ = 7.46, *p* = 0.031). Carbohydrate intake that is too low is undoubtedly linked with dietary fibre intake that is too low among female athletes when compared with male athletes (20.1% vs. 66.1%, *p* < 0.001).

Athletes compensate for the carbohydrate deficiency by consuming more than the recommended amount of fat. Regardless of sport or gender, average fat energy value of majority athletes (76.5%) exceeds 35% of calories. Therefore, 87.9% of athletes consume too high a level of SFA and 78.5% consume too much cholesterol. Increased SFA and cholesterol in the diet is characteristic of male athletes compared to female athletes (87.0% vs. 53.2%, *p* < 0.001). In contrast, 52.6% of athletes consume too low a level of PUFA. Almost all Lithuanian elite athletes (99.5%) are low in omega-3 FA, while 41.3% of athletes consume an insufficient level of omega-6 FA in their food.

The mean dietary intake of protein ranged from 1.7 ± 0.7 g/kg of body mass per day to 1.8 ± 0.6 g/kg of body mass per day among male athletes and 1.2 ± 0.4 g/kg of body mass per day to 1.5 ± 0.7 g/kg of body mass per day among female athletes. A more detailed analysis of the survey data has shown that 38.1% of athletes consume too little protein, and 29.1% consume too much. Only 32.8% of athletes consume the recommended amount of protein. Furthermore, we found that protein consumption depends on gender differences. Female athletes (61.3%) consume too little protein, while male athletes (36.2%) consume too much (χ^2^ = 19.29, *p* < 0.001).

The compliance with RDI [26] of vitamins and minerals consumed by athletes was analysed (Table 2 and Table 3). It was found that athletes consume significantly (*p* < 0.01) more vitamins A, E, B_1_, B_2_, B_3_, B_6_, B_9_, B_12_, C, and minerals potassium, magnesium, phosphorus, manganese, copper and zinc compared to RDI.

According to the RDI, women should consume at least 15 mg of iron per day. The data of our study confirm that the amount of iron (18.8 ± 7.5 mg/day) in the diet of female athletes is higher than the RDI (*p* < 0.001). However, iron requirements for all female athletes may be increased by up to 70% of the estimated average requirement [5,24]. Therefore, the actual iron intake of female athletes compared to the recommended one (25 mg/day), the iron deficiency in the diet corresponds to −6.2 ± 7.5 mg/day (95% Confidence interval (CI): −8.1, −4.3, *p* < 0.001).

In the diets of male and female athletes we also found lower levels of vitamin D than RDI: 144 ± 104 International Unit (IU)/day and 88 ± 76 IU/day, respectively. A statistically significant differences (*p* < 0.01) between vitamin D intake and RDI in the male and female groups were −256 ± 104 IU/day (95% CI: −272, −240) and −312 ± 76 IU/day (95% CI: −328, −292), respectively.

According to scientific advice, calcium intakes of 1500 mg/day and 1500–2000 IU of vitamin D are needed to optimize bone health in athletes [25]. In our study, both male and female athletes were deficient in calcium in their diets at −272 ± 511 mg/day (95% CI: −347, −199, *p* < 0.001) and −527 ± 472 mg/day (95% CI: −693, −452, *p* < 0.001), respectively.

### 3.2. Dietary Supplementation

Most athletes taking dietary supplements usually choose carbohydrates (86%), vitamins (81.3%), minerals (74.5%), protein (70.4%) and multivitamins (61.8%). More rarely do they choose caffeine (36%), omega-FA (46.7%), creatine (24.6%), carnitine (25.4%) or herbal supplements (19%).

Our study evaluated the duration of supplementation use. We determined that 28–37% of the respondents consume carbohydrate, protein, vitamin, mineral and multivitamin supplements for 5–8 months a year or more, while 38–49% of athletes for 1–4 months a year. Meanwhile, elite athletes use omega-FA, creatine, carnitine, caffeine and herbal dietary supplements both less frequently and for shorter periods. Only 14–20% of athletes use the aforementioned supplements for no longer than 1–4 months per year.

We searched for the links between sports, sex and supplementation. It is noteworthy that athletes who train for prolonged endurance often use caffeine during the training period. Meanwhile, there are almost no anaerobic sports athletes who use caffeine in training (only 3.8%) compared with prolonged endurance athletes (20.5%) and mixed aerobic and anaerobic sports athletes (19.6%) (*p* = 0.011). After evaluating the use of food supplements by male and female athletes, we established that during the precompetition period, protein and caffeine food supplements were consumed more by male athletes (68.2% and 20.5%) than female (50% and 8.3%) (*p* = 0.013 and *p* = 0.032). We found that female athletes use more herbal food supplements than males during the training period (26.7% vs. 14.8%, *p* = 0.050) (Table 4).

### 3.3. Interactions between Dietary Intake, Food Supplementation and Body Composition

According to the survey, body weight and BMI of all elite athletes ranges within the norms. Male athletes’ BF is considered acceptable, and LBM is optimal. In contrast, BF readings for females in different sports were distributed unevenly. We determined that BF readings were too high for anaerobic sports athletes, representing 27.1 ± 5.9% of body weight. Meanwhile, female athletes training for mixed aerobic and anaerobic sports and aerobic sports have BF that accounts for, respectively, 22.5 ± 3.6% and 22.0 ± 3.7% of total body weight, which is optimal.

We established that the improved physical development is due to the increased muscle mass among mixed aerobic and anaerobic sports athletes (MFMI of 5.7 ± 2.8), and male athletes training for endurance (MFMI of 5.2 ± 2.8). Similarly, female athletes in those sports had an average MFMI of 3.3 ± 0.7 and 3.4 ± 0.9,, respectively, as well as a low MFMI in those in anaerobic sports (2.4 ± 0.6), meaning the insufficient physical development is determined by a relatively excessive BF for anaerobic sports female athletes (Table 5).

Optimum body composition of athletes basically depends on LBM and BF masses. Surmountable physical stress and diet have a great influence on the latter. According to the survey, regardless of sport, sex and age, elite athlete LBM (%) is not determined by complex nutritional supplements consisting of carbohydrates, proteins, creatine, vitamins, minerals, multivitamins and omega-3 FA. There was no statistically significant difference between the LBM of those taking and not taking supplements (%) (Table 6).

On the other hand, the analysis of the survey data using the multivariate regression method shows a 3.5% increase in LBM (%) for the Lithuanian elite athletes tested during precompetition and workouts, which only helps increased protein in the usual diet (*p* = 0.057). Meanwhile, neither food ration energy value nor carbohydrate content encourages LBM (%) development (Table 7).

Furthermore, we did not established significant effects on athlete BF (%) after evaluating consumption of food supplements (carbohydrates, proteins, creatine, caffeine, herbal supplements, multivitamins, vitamins, minerals and omega-3 FA), except for the fact that athletes with a BF (%) higher than 1.6% take carnitine supplements more frequently (Table 8).

## 4. Discussion

The research revealed flawed features of elite athlete nutrition. More than 60% of the athletes tested consume insufficient dietary carbohydrates. Other scholars have also found a lack of carbohydrates in the nutrition of athletes [32,33,34,35,36,37].

38% of the athletes we surveyed had diets low in protein. Protein intake was above recommendations in 33% of the athletes. In terms of the nutritional differences between different subsets of athletes, we can characterise female athletes as protein deficient. A more detailed analysis of the survey data showed that Lithuanian male athletes consume too much protein. Other researchers did not find too little protein in the nutrition of athletes. Meanwhile Mars cal-Arcas, et al. [38], Kim H, et al. [39], and other researchers [40,41] indicate excessive protein content in the nutrition of athletes. According to scientific data the use of high amounts of protein in competitive sports may be necessary and useful. The consumption of 1.4–2.0 g/kg/day of protein is reasonable for professional athletes training for both strength and endurance [8].

Almost all of our athletes tested high for fat and saturated FA. Other researchers have indicated the same findings also [38]. There is no scientific evidence that high-fat food intake results in overweight athletes or in increased cholesterol [41]. Excess fat diets have a connection to homocysteine levels in the body, promoting the emergence of cardiovascular diseases [42]. The problem with a diet too rich in fat is the lack of carbohydrates, which can slow down the adaptation to physical loads [43] weaken the immune system, and increase the risk of sports injuries [6]. With insufficient levels of restored endogenous glycogen storage, muscles require increased effort from the central nervous system to alleviate physical stress, and there is a risk factor for overtraining [6]. However, in our study we did find athletes having diets with excess cholesterol. Meanwhile, other researchers point out that cholesterol in the diet of athletes does not exceed the recommended amount [37,44].

The diets of almost all elite Lithuanian athletes are deficient in vitamin D. In other countries, the lack of this vitamin in sports nutrition is regarded as endemic [45,46]. Vitamin D deficiency is common in Europe, mostly in countries of Northern European latitudes (>35° N) such as the UK, Ireland, Denmark, France, Germany, and Lithuania [47,48]. Since Lithuanian high-performance athletes tend to consume little vitamin D from the diet and dietary interventions alone have not been shown to be a reliable means to resolve insufficient status, supplementation above the current RDI and/or responsible UVB exposure may be required to maintain sufficient vitamin D status.

Most of the elite Lithuanian athletes (96%) consume dietary supplements. A similar number of athletes take the same level of dietary supplements in the United States, Sri Lanka, Italy and Australia, whereas in other countries, there are fewer users [11,12,49,50,51,52,53]. Most elite Lithuanian athletes (over 70%) consume carbohydrate, vitamin and mineral dietary supplements. There are fewer athletes taking other supplements (omega-3 FA, creatine, carnitine, caffeine). In addition, the athletes we tested consume these dietary supplements for a shorter period of time than the athletes in other countries. Supplementation differences depending on gender and sport are not very significant, except that males more frequently consume protein supplements compared with females, who prefer herbal supplements.

Researchers have assessed the consumption of dietary supplements (carbohydrate drinks, gels, bars and protein) corresponding to actual diet [5,31]. In our study, most of the diets of athletes lack carbohydrates, and therefore carbohydrate food supplements are appropriate. Meanwhile, protein supplements are exclusively used by male athletes (68%) more than by females (50%). This can be appropriate only for female athletes, because their diet is low in protein. On the other hand, the use of protein supplements should be more compatible with the actual diet and exercises workload plans among male athletes.

Medical supplementation (multivitamins; the minerals like iron and calcium; vitamins, particularly vitamin D; omega-3 FA) is recommended only in case of deficiencies in these substances and only with medical supervision [5,31]. Meanwhile, both our research and other scientific studies show that athletes most frequently consume multivitamin, vitamin and mineral dietary supplements and use them at their discretion [9,10,49]. Female athletes in Lithuania are especially recommended to use additional amounts of calcium and vitamin D for the prevention of osteoporosis [21,54]. Lithuanian female athletes tend to consume too little iron. Iron deficiency is associated with female athletes’ inadequate nutrition, anaemia so they are recommended to consume 70% more than the RDI of iron [54]. Iron deficiency, with or without anaemia, can impair muscle function and limit work capacity leading to compromised training adaptation and athletic performance [55].

On the other hand, we discovered a deficit of omega-3 FA in the diet 99% of the Lithuanian athletes, and the athletes rarely take omega-3 FA supplements. Therefore, omega-3 FA supplementation offers should be increased, as well as the demand among the athlete segment in Lithuania.

The athlete’s training process, proper nutrition and an adequate use of dietary supplements are important in not only ensuring the adaptation to physical loads but also in acquiring the optimal body composition. Building LBM and maintaining relatively low BF are some of the most important tasks of physical training. We assessed LBM (%) and BF (%) dependence on diet and nutritional supplements and found that Lithuanian professional athletes have a higher level of LBM (3.5% (95% CI: −0.107, 7.070)) depending on the increased use of dietary protein. Meanwhile, nutritional supplement mixes (carbohydrates, proteins, creatine, multivitamins and minerals, omega-3 FA) do not have associations with the development of LBM (%). This may be associated with the effects of supplements being so low compared with the conventional diet that the latter was not even identified. We did not find a direct influence of creatine monohydrate use stimulating muscle hypertrophy, because its duration is short (1–4 month/year) and the number of users is less than 20%. We may conclude that the optimal LBM (%) of male athletes develops during the training process and when using a quantity of protein equivalent to 1.7–1.8 g/kg/day. In contrast, the LBM (%) of female athletes is too small for their intake of protein of 1.2–1.5 g/kg/day and is sufficient to maximally stimulate muscular hypertrophy. Due to the fact that the energy intake of female athletes is lower than EER, their intake of protein should be higher.

The strength of our study is that it has examined the actual nutrition and body composition of almost all Lithuanian high-performance athletes. The study of actual nutrition is much more objective compared to the study of eating habits. Actual nutrition data combined with objective indicators of athletes’ body composition allow us to predict and implement targeted measures and recommendations for optimizing athletes’ nutrition for the next Olympic cycle. The data on the dietary supplement consumption habits provide opportunities to rationalize the diet of athletes and to better combine the dietary supplements taken according to athletic goals, body composition, and actual nutrition. The data and recommendations of our study can be applied in practice by including them in the current sportsmen training programmes of the Lithuanian National Olympic Committee (LNOC): Tokyo 2020 (the Olympics of the summer of 2020 has been moved to 2021) and Beijing 2022.

In the future, continuous investigation and monitoring of body composition and actual nutrition should be carried out during each 4-year Olympic cycle. The weakness of our study is that it was only a 24 h food recall survey of actual nutrition during the precompetition period. Therefore, in the future, in cooperation with the Lithuanian Sports Medicine Center (LSMC), it is necessary to monitor the actual nutrition and other health indicators of highly skilled athletes for a period from three to seven days during the preparatory and competition periods.

## 5. Conclusions

The main nutrients in the diets of the athletes researched are unbalanced (carbohydrates, proteins and fats). Elite athletes consume insufficient amounts of carbohydrates, vitamin D, calcium, polyunsaturated fatty acids, omega-3 and omega-6 fatty acids, while their diet is too rich in fat, saturated fatty acids, cholesterol, and they use proteins irrationally. Sport nutritionists should also focus on the risk of malnutrition in female athletes. In this case, the diet of female athletes is exceptionally deficient in energy, carbohydrates, dietary fibre, protein, iron, calcium and vitamin D.

The carbohydrates, vitamins, minerals, protein supplements and multivitamins rationally used by athletes, tend to partially compensate for the lack of essential nutritional elements (macronutrients and micronutrients) in their diet. Meanwhile, nutritional supplement mixes (carbohydrates, proteins, creatine, multivitamins and minerals, omega-3 fatty acids) do not have associations with the development of lean body mass (%), excluding the fact that Lithuanian professional athletes have a higher level of lean body mass (3.5% (95% CI: −0.107, 7.070)) depending on the increased intake of dietary protein. It can be stated that the effect of food supplements on the body composition of athletes is too small compared to the normal diet.

The training of athletes in different sports needs to be very individualised in terms of the mesocycling of sports goals, including the changes in body composition, giving priority to the formation of eating habits and then, dietary supplementation.

## Figures and Tables

**Table 1 medicina-56-00247-t001:** Energy intake, estimated energy requirement and macronutrient intake by athletes depending on sports and gender.

Energy and Nutrients	Anaerobic		Mixed Aerobic and Anaerobic		Aerobic	
	M	F	M	F	M	F
EER (kcal)	4845 ± 1118	4001 ± 582	4635 ± 1058	3643 ± 529	4629 ± 842	3461 ± 660
EER (kcal/kg)	56.8 ± 8.6	53.2 ± 8.7	64.1 ± 7.8	62.6 ± 9.3	62.1 ± 8.6	59.5 ± 8.2
EI (kcal/day)	4244 ± 1389	2197 ± 888	3299 ± 857	2275 ± 646	3752 ± 825	2484 ± 1065
EI (kcal/kg)	50.5 ± 16.0	30.1 ± 15.1	47.3 ± 15.3	39.8 ± 13.3	51.0 ± 12.5	43.3 ± 19.8
CHO (g/kg)	5.7 ± 2.0	3.7 ± 2.2	5.3 ± 1.9	4.7 ± 1.5	5.8 ± 1.9	5.0 ± 2.4
Dietary fibre (g)	43.4 ± 18.4	29.0 ± 10	31.2 ± 11.6	22.3 ± 6.3	36.0 ± 14.2	24.0 ± 12.9
Fat %	40.4 ± 6.9	35.9 ± 9.4	40.0 ± 7.4	37.1 ± 7.9	40.9 ± 8.1	39.4 ± 7.3
SFA %	13.9 ± 3.7	12.6 ± 3.8	13.4 ± 3.4	14.6 ± 4.0	14.9 ± 2.5	13.0 ± 3.0
PUFA %	6.5 ± 2.7	5.5 ± 2.0	6.3 ± 1.8	4.9 ± 1.4	6.1 ± 11.7	5.3 ± 1.2
Omega-3 FA%	0.3 ± 0.2	0.3 ± 0.1	0.3 ± 0.1	0.3 ± 0.1	0.3 ± 0.1	0.4 ± 0.1
Omega-6 FA %	6.0 ± 2.6	5.0 ± 2.1	5.9 ± 1.8	4.2 ± 0.4	5.6 ± 1.7	4.8 ± 1.3
Cholesterol (g)	1.1 ± 0.5	0.8 ± 0.6	0.9 ± 0.5	0.6 ± 0.4	0.9 ± 0.4	0.5 ± 0.3
PRO (g/kg)	1.8 ± 0.6	1.2 ± 0.4	1.7 ± 0.7	1.4 ± 0.6	1.8 ± 0.5	1.5 ± 0.7

M—male; F—female; PRO—protein; EI—energy intake; EER—total estimated energy requirement; PRO—protein; CHO—carbohydrates; FA—fatty acids; SFA—saturated fatty acids; PUFA—polyunsaturated fatty acids.

**Table 2 medicina-56-00247-t002:** Vitamin intake by athletes depending on gender.

Vitamins	Gender	Intake	RDI	Delta Intake (Actual − RDI)	*p*
A (µg RE ^1^)	M	1224.6 ± 557.0	900	324.6 ± 557.0 (243.8, 405.4)	<0.001
	F	1000.5 ± 1117.7	700	300.5 ± 1117.7 (16.6, 584.3)	0.04
D (IU)	M	144 ± 104	400	−256 ± 104 (−272, −240)	<0.001
	F	88 ± 76	400	−312 ± 76 (−328, −292)	<0.001
E (mg a-TE ^2^)	M	25.5 ± 10.8	12	13.5 ± 10.8 (12.0, 15.1)	<0.001
	F	14.1 ± 6.5	10	4.1 ± 6.5 (2.5, 5.7)	<0.001
B_1_ (mg)	M	2.4 ± 1.3	1.4	1.0 ± 1.3 (0.8, 1.2)	<0.001
	F	1.5 ± 0.9	1.1	0.4 ± 0.9 (0.2, 0.6)	<0.001
B_2_ (mg)	M	3.2 ± 1.3	1.6	1.6 ± 1.3 (1.4, 1.8)	<0.001
	F	2.1 ± 1.1	1.3	0.8 ± 1.1 (0.5, 1.0)	<0.001
B_3_ (mg NE ^3^)	M	31.5 ± 14.2	19	12.5 ± 14.2 (10.4, 14.6)	<0.001
	F	19.6 ± 8.1	15	4.6 ± 8.1 (2.5, 6.7)	<0.001
B_6_ (mg)	M	4.1 ± 1.4	1.6	2.5 ± 1.4 (2.3, 2.7)	<0.001
	F	2.6 ± 1.0	1.3	1.3 ± 1.0 (1.0, 1.5)	<0.001
B_9_ (µg)	M	287.9 ± 116.5	200	87.9 ± 116.5 (71.0, 104.8)	<0.001
	F	204.8 ± 91.6	200	4.8 ± 91.6 (−18.4, 28.1)	0.68
B_12_ (µg)	M	5.8 ± 2.8	3	2.8 ± 2.8 (2.4, 3.2)	<0.001
	F	4.1 ± 4.5	3	1.1 ± 4.5 (−0.1, 2.2)	0.07
C (mg)	M	144.4 ± 95.8	80	64.4 ± 95.8 (50.5, 78.3)	<0.001
	F	134.5 ± 115.7	80	54.5 ± 115.7 (25.2, 83.9)	<0.001

Data is normally distributed and presented as means ± standard deviation (SD) with the 95% confidence interval (CI) in parentheses. Delta Intake = Actual intake − Recommended intake. *p*-value from paired samples *t*-test (normally distributed data). RDI—Recommended Dietary Intake [26]. ^1^—RE (retinol equivalent); ^2^—TE (α-tocopherol equivalent); ^3^—NE (niacin equivalent).

**Table 3 medicina-56-00247-t003:** Mineral intake by athletes depending on gender.

Minerals	Gender	Intake	RDI	Delta Intake (Actual − RDI)	*p*
Potassium (mg)	M	5361.4 ± 2085.4	3500	1861.4 ± 2085.4 (1558.9, 2163.9)	<0.001
	F	3598.6 ± 1532.2	3100	498.6 ± 1532.2 (109.5, 887.7)	0.01
Calcium (mg)	M	1227 ± 511	1500 ^2^	−272 ± 511 (−347, −199)	<0.001
	F	927 ± 474	1500 ^2^	−527 ± 472 (−693, −452)	<0.001
Magnesium (mg)	M	533.8 ± 227.2	350	183.8 ± 227.2 (150.9, 216.8)	<0.001
	F	354.3 ± 147.9	300	54.3 ± 147.9 (16.8, 91.9)	0.01
Phosphorus (mg)	M	2076.7 ± 648.6	700	1376.7 ± 648.6 (1282.6, 1470.8)	<0.001
	F	1356.3 ± 524.8	700	656.3 ± 524.8 (523.0, 789.6)	<0.001
Iron (mg)	M	28.8 ± 9.8	10	18.8 ± 9.8 (17.4, 20.2)	<0.001
	F	18.8 ± 7.5	15/25 ^1^	3.8 ± 7.5 (1.9, 5.7), −6.2 ± 7.5 (−8.1, −4.3) ^1^	<0.001
Manganese (mg)	M	5.8 ± 2.6	3	2.8 ± 2.6 (2.5, 3.2)	<0.001
	F	4.3 ± 2.7	3	1.3 ± 2.7 (0.6, 2.0)	<0.001
Copper (mg)	M	2.8 ± 1.0	1	1.8 ± 1.0 (1.6, 1.9)	<0.001
	F	2.0 ± 0.8	1	1.0 ± 0.8 (0.8, 1.2)	<0.001
Zinc (mg)	M	17.8 ± 5.8	10	7.8 ± 5.8 (6.9, 8.6)	<0.001
	F	10.4 ± 3.6	10	0.4 ± 3.6 (−0.5, 1.3)	0.40

^1,2^—Recommended values are derived through a combination of published review articles [5,24,31] and clinical experience.

**Table 4 medicina-56-00247-t004:** Dietary supplement use during the preparatory period depending on athlete gender.

Dietary Supplements	Male, % (N)	Female, % (N)	*p*
Carbohydrates	75.0 (132)	75.0 (45)	0.52
Amino acids	68.2 (120)	50.0 (30)	0.01
Vitamins	74.4 (131)	81.7 (49)	0.30
Minerals	67.6 (119)	73.3 (44)	0.52
Multivitamins	57.4 (101)	51.7 (31)	0.46
PUFA	42.0 (74)	51.7 (31)	0.23
Creatine	21.0 (37)	23.3 (14)	0.72
Carnitine	21.6 (38)	30.0 (18)	0.22
Caffeine	20.5 (36)	8.3 (5)	0.03
Herbal	14.8 (26)	26.7 (16)	0.05

**Table 5 medicina-56-00247-t005:** Anthropometric data of athletes.

Factor	Gender	Anaerobic	Mixed Aerobic and Anaerobic	Aerobic
Height (cm)	M	193.2 ± 11.8	176.2 ± 10.9	183.5 ± 8.9
	F	178.9 ± 3.4	167.1 ± 8.0	167.9 ± 6.2
BM (kg)	M	85.1 ± 14.3	72.9 ± 17.2	74.9 ± 11.4
	F	76.4 ± 15.3	58.8 ± 7.9	58.3 ± 8.2
LBM (kg)	M	69.1 ± 8.5	60.6 ± 11.9	62.1 ± 7.6
	F	55 ± 5.9	45.4 ± 5.1	45.3 ± 5.3
LBM (%)	M	81.8 ± 4.5	84.0 ± 5.5	83.4 ± 4.1
	F	72.9 ± 6.0	77.1 ± 3.4	77.9 ± 3.9
BF (kg)	M	16.1 ± 7.0	12.4 ± 6.5	12.8 ± 4.5
	F	22.9 ± 8.7	13.4 ± 3.6	13.0 ± 3.5
BF (%)	M	18.2 ± 4.5	16.0 ± 5.4	16.6 ± 4.1
	F	27.1 ± 5.9	22.5 ± 3.6	22.0 ± 3.7
BMI (kg/m^2^)	M	22.9 ± 3.7	23.0 ± 3.5	22.1 ± 2.0
	F	23.6 ± 4.2	21.0 ± 1.8	20.6 ± 2.0
MFMI	M	4.5 ± 1.4	5.7 ± 2.8	5.2 ± 2.8
	F	2.4 ± 0.6	3.3 ± 0.73	3.4 ± 0.9

M—male; F—female. Data is normally distributed and presented as means ± standard deviation (SD). LBM—lean body mass; BM—body mass; BF—body fat; BMI—body mass index; MFMI—muscle and fat mass index

**Table 6 medicina-56-00247-t006:** Relationship between food supplementation and athlete LBM (%).

LBM (%)	β	95% CI	*p*
Carbohydrates (drinks, sport bars, gels)	−0.361	(−2.030, 1.309)	0.67
Protein (blends, EAA)	−0.660	(−2.017, 0.696)	0.34
Creatine	−0.675	(−2.067, 0.716)	0.34
Vitamins	0.399	(−1.213, 2.001)	0.63
Minerals	0.412	(−1.045, 1.871)	0.58
Multivitamins	0.530	(−0.708, 1.768)	0.40
Omega-3 FA	0.037	(−1.122, 1.197)	0.95

EAA—essential amino acids; food supplementation influence on LBM (%) is estimated controlling for athlete sport, gender and age (adjusted for sports type, gender, and age). F(11, 224) = 9.72, *p* < 0.0001, R^2^ = 0.32.

**Table 7 medicina-56-00247-t007:** Relationship between dietary intake and athlete LBM (%).

LBM (%)	β	95% CI	*p*
Protein (g/kg) (ln)	3.481	(−0.107, 7.070)	0.06
Carbohydrates (g/kg) (ln)	4.861	(−1.639, 11.361)	0.14
Fat (g/kg) (ln)	3.271	(−1.905, 8.446)	0.21
Energy (kcal/kg) (ln)	−5.748	(−20.120, 8.623)	0.43

Eating habit influence on LBM (%) is estimated controlling for athlete sport, gender and age (adjusted for sports type, gender, and age). F(8, 238) = 26.30, *p* < 0.0001, R^2^ = 0.45.

**Table 8 medicina-56-00247-t008:** Relationship between food supplementation and athlete BF (%).

BF (%)	β	95% CI	*p*
Carbohydrates (drinks, sport bars, gels)	−0.252	(−1.574, 1.071)	0.71
Protein (blends, EAA)	0.337	(−0.931, 1.606)	0.60
Creatine	0.943	(−0.518, 2.404)	0.21
Carnitine	1.570	(0.161, 2.979)	0.03
Caffeine	−0.410	(−1.927, 1.106)	0.59
Herbal	0.780	(−0.780, 2.340)	0.33
Multivitamins	−0.668	(−1.870, 0.534)	0.28
Vitamins	−0.414	(−1.904, 1.077)	0.59
Minerals	−0.294	(−1.654, 1.066)	0.67
Omega-3 FA	0.015	(−1.146, 1.176)	0.98

Food supplementation influence on BF (%) is estimated controlling for athlete sport, gender and age (adjusted for sports type, gender, and age). F(14, 221) = 8.11, *p* < 0.0001, R^2^ = 0.34.

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
