# Peer review of "Actual Nutrition and Dietary Supplementation in Lithuanian Elite Athletes"

_medicina, 2020, doi:10.3390/medicina56050247_

Round 1

Reviewer 1 Report

The authors fully answered all the questions asked. The article meets the criteria for publication in "MEDICINA"

Author Response

Thank you for the quick review of the article. We have taken all reviewers’ comments into consideration and submit a revised article.

Reviewer 2 Report

The work is interesting, commnents can be found in the attachment. The value of the work has been enhanced by the earlier added (and highlighted) corrections. 

Author Response

Thank you for the quick review of the article. We have taken all reviewers’ comments into consideration and submit a revised article. Also, in this letter, we provide our responses to all the reviewers' observations.

Responses to the reviewers’ observations

Reviewer’s observation: “Within this context, it is worth considering changing the title, because in the current version, it does not fully reflect the content of this work. All the more that the authors did not evaluate eating habits but rather the way of eating (supply of energy and nutrients).”

Reply and corrections. Following the reviewer's observation, we've changed the name of the manuscript to a more specific one: “Actual Nutrition and Dietary Supplementation in Lithuanian Elite Athletes”. We have also clarified the objective of the study as follows: “Study objective: to characterize the actual nutrition and dietary supplements taken by elite Lithuanian athletes and to identify the relationship between the dietary intake, dietary supplementation and body composition of elite athletes in training processes.”

Reviewer’s observation: It should be highlighted that the authors rightly emphasize that the basis for sports nutrition should be a diverse and balanced diet, and not dietary supplements.

Reply and corrections.  In the article, we supplemented the conclusions section and the abstract according to the reviewer's observations: “Meanwhile, nutritional supplement mixes (carbohydrates, proteins, creatine, multivitamins and minerals, omega-3 FA) do not have associations with development of LBM (%)). Except that Lithuanian professional athletes have a higher level of LBM (3.5% (95% CI: -0.107, 7.070) depending on increased intake of dietary protein. It can be stated that the effect of food supplements on the body composition of athletes is too small compared to the normal diet.”

Reviewer’s observation: The ‘Abstract’ was prepared in a synthetic way. This comment concerns the ‘Material and Methods’ section, in which the authors indicated methods for assessing diet (24-hour interview) and supplementation (author's questionnaire), while omitting the methods of measuring somatic indices (requires supplementation).

Reply and corrections. Taking into account the observations, we improved the Abstract: “Measurements of body composition were performed using the BIA (bioelectrical impedance analysis) tetra-polar electrodes and measuring resistivity with 8-12 tangent electrodes at different frequencies of signal: 5, 50 and 250 kHz.”

Reviewer’s observation: The ‘Keywords’ do not fully relate to the scope of work. I suggest their supplementation (e.g. diet, body composition indices, etc.).

Reply and corrections. The keywords were supplemented.Keywords: elite athletes; actual nutrition; diet; dietary supplements; body composition.” 

Reviewer’s observation: The ‘Discussion’ part is technically correct. I propose the authors prepare an additional paragraph of the ‘Discussion’, in which they would indicate the strengths and weaknesses of their work and directions for further research.

Reply and corrections. The strength of our study is that it has examined the actual nutrition and body composition of almost all Lithuanian high-performance athletes. The study of actual nutrition is much more objective compared to the study of eating habits. Actual nutrition data combined with objective indicators of athletes’ body composition allow to predict and implement targeted measures and recommendations for optimizing athletes' nutrition for the next Olympic cycle. Data on consumption habits of dietary supplements provide opportunities to rationalize the diet of athletes and to better combine the dietary supplements taken taking into account the athletic goals, body composition, and actual nutrition. The data and recommendations of our study can be applied in practice by including them in the current athlete training programmes of the Lithuanian National Olympic Committee (LNOC): “Tokyo 2020” (The Olympics of the summer of 2020 has been moved to 2021) and “Beijing 2022”. In the future, continuous investigation and monitoring of body composition and actual nutrition should be carried out during each 4-year Olympic cycle. The weakness of our study is that it was only 24-hour recall survey of actual nutrition during the pre-competition period. Therefore, in the future, in cooperation with the Lithuanian Sports Medicine Center (LSMC), it is necessary to monitor the actual nutrition and other health indicators of highly skilled athletes for a period from 3 to 7 days during the preparatory and competition periods.  

Reviewer’s observation: The formulated ‘Conclusions’ are justified. Nonetheless, it seems that there are no conclusions concerning the relationships between the analysed variables (diet, supplementation, somatic indices).

Reply and corrections.  We have supplemented the conclusions section according to recommendations: “Nutritional supplement mixes (carbohydrates, proteins, creatine, multivitamins and minerals, omega-3 FA) do not have associations with development of LBM (%). Except that Lithuanian professional athletes have a higher level of LBM (3.5% (95% CI: -0.107, 7.070)) depending on increased intake of dietary protein.”

Reviewer’s observation: Moderate English changes required.

Reply and corrections.  We have corrected and improved the English language of the article.

Sincerely,

Marius Baranauskas

This manuscript is a resubmission of an earlier submission. The following is a list of the peer review reports and author responses from that submission. Reviewer 1 did not agree to revise again. Reviewer 2 in first round became reviewer 1 in the latest submission. Reviewer 2 in latest submission in a new independent reviewer. 

Round 1

Reviewer 1 Report

Review Report

Brief summary

This paper was written to “characterize the eating habits and food supplements taken by elite Lithuanian athletes”. The authors also claimed they could “identify the effects of different nutrients on body composition”. However, the study design and methodology employed do not allow such conclusions to be made.

Broad comments highlighting areas of strength and weakness. These comments should be specific enough for authors to be able to respond.

This study has done extremely well to undertake any sort of analysis on such a large number (n = 247) of elite athletes. However, the research questions are not well defined and this has led to the inclusion of excessive amounts of data/comparisons that do not help answer clearly defined questions. Most importantly, the methods are severely limited and do not allow the authors to support their conclusions. Neither can the “effects of different nutrients on body composition” be ascertained from a correlative analysis, nor has sufficient thought been given to the limitations of 24 h dietary recall on the assessment of athletes’ intake. As athletes were from a large range of sports, doing different types of training, it is hard to understand how the 24 h “snapshot” taken of these athletes’ diets could have been taken to represent a typical day’s training in the “preparatory period” for all athletes. Insufficient evidence is provided to support the use of validated instruments (e.g. the neither the supplement questionnaire, nor portion-size resource are properly referenced), while the limitations to other methods (24 h dietary recall/bioelectrical impedance analysis) have not been discussed or overtly addressed.  

Specific comments

Providing a complete list of specific issues will not be helpful, considering the flaws in design and methodology. In summary:

  • A cross-sectional, observational study cannot be used to draw conclusions on cause and effect
  • The limitations of the methods of dietary analyses and body composition assessment used prevent robust conclusions being made on these athletes’ diets, and have not even been mentioned, yet alone addressed
    • Furthermore, conclusions are made on athletes’ vitamin D status from only dietary analysis, which is not valid
  • Many inaccurate statements and references have been used
    • g. whilst European RNI values for n3/n6 fats are debateable anyway, the reference provided (ACSM 2016) doesn’t even provide such recommendations for PUFAs
    • g. The sentence “Incomplete restoration of glycogen storage in the muscles and liver between workouts requires more central nervous system effort to overcome high-intensity physical loads” is not supported with evidence (lines 40 – 41)
  • Inadequate use of scientific English makes the study hard to follow
    • For example, when discussing the sub analysis on athlete protein intake, the authors switch between describing a “3.5% increase in LBM (%)” (line 323), to “3.5% of Lithuanian professional athletes” (line 433).
    • From the data provided, it would seem they are referring to the regression analysis and inferring a 3.5 % greater amount of lean mass. However, the wording on line 323 sounds like they have tested at multiple time-points and observed an increase in LBM, whilst the wording on line 433 makes the 3.5 % seem like a subgroup of athletes.
    • In either case, the findings were not significant and have therefore been incorrectly interpreted.
  • Numerous instances of incorrect/inappropriate English terminology further undermine the authors’ points;
    • g. nutrition is “determined by the state of one’s health” (line 11), meaning that individuals/athletes will eat differently if they are more or less healthy? Why is this relevant?
    • blended (aerobic and anaerobic) force” (line 168) is incorrect, as it is power that is the application of force in a given time, and the word blended doesn’t work in this context (far too “physical”, suggesting these two components are tangible, physical objects that have been mashed and mixed.

Originality/Novelty: Is the question original and well defined? Do the results provide an advance in current knowledge?

Significance: Are the results interpreted appropriately? Are they significant? Are all conclusions justified and supported by the results? Are hypotheses and speculations carefully identified as such?

Quality of Presentation: Is the article written in an appropriate way? Are the data and analyses presented appropriately? Are the highest standards for presentation of the results used?

Scientific Soundness: is the study correctly designed and technically sound? Are the analyses performed with the highest technical standards? Are the data robust enough to draw the conclusions? Are the methods, tools, software, and reagents described with sufficient details to allow another researcher to reproduce the results?

  • The research questions are not well defined and this has led to the inclusion of excessive amounts of data/comparisons that do not help answer clearly defined questions. Most importantly, the methods are severely limited and do not allow the authors to support their conclusions
  • A cross-sectional, observational study cannot be used to draw conclusions on cause and effect
  • The limitations of the methods of dietary analyses and body composition assessment used prevent robust conclusions being made on these athletes’ diets, and have not even been mentioned, yet alone addressed
    • Furthermore, conclusions are made on athletes’ vitamin D status from only dietary analysis, which is not valid

Interest to the Readers: Are the conclusions interesting for the readership of the Journal? Will the paper attract a wide readership, or be of interest only to a limited number of people? (please see the Aims and Scope of the journal)

Overall Merit: Is there an overall benefit to publishing this work? Does the work provide an advance towards the current knowledge? Do the authors have addressed an important long-standing question with smart experiments?

  • Not relevant as study is unsound

English Level: Is the English language appropriate and understandable?

  • Inadequate

Manuscripts submitted to MDPI journals should meet the highest standards of publication ethics:

Manuscripts should only report results that have not been submitted or published before, even in part.

Manuscripts must be original and should not reuse text from another source without appropriate citation.

  • No reason to doubt ethics or originality

Overall Recommendation

Please provide an overall recommendation for the publication of the manuscript as follows:

  • Reject: The article has serious flaws, makes no original contribution, and the paper is rejected with no offer of resubmission to the journal.

Reviewer 2 Report

Links between the eating habits and food supplements use of elite athletes

The paper is globally well written, the reviewer has some comments that need to be solved before considering it for a publication. Comments are specifically reported here below:

Abstract:

Line 16: 76.7% of what? Total Lituanian athletes?

Introduction

Line 33: Nutrition is partly determined by health, or health is partly determined by nutrition? Please, comment.

Line 36: What do authors mean with ‘only then’?

Line 44: Not only the professional athletes need to reach the RDI. Maybe it would be the case to say that professional athletes, due to high workloads, have different RDI respect to “normal people”? Please, Comment

Methods

Line 67: In which period of the season data have been collected? Body composition and dietary habits of the athletes may vary in function of the period (eg. off-season, preparatory, competitive). Furthermore, if data were collected between 2016 and 2017, it was a period following the Rio Olympics: could have this affected the results? Authors are suggested to discuss this point.

Line 95: Was the weight of the muscle obtained by the BIA? Please, Report method used.

Line 106 Atlas foodstuffs: Please, report reference

Line 122: “…meets the prescribed RDI for a given country”. It is unclear. Please, clarify

Line 145: Validated by…?

Line 158: Consent to participate? In particular for <18 yo athletes.

Results

Line 165: Find a synonimum, please.

Line 165: Why did you split into two age groups? by what criteria?

Line 167: This has already been reported in the Methods section. Please remove one of the two, to avoid repetitions.

Line 177-178: This is a discussion phrase. Please, move in discussion

Line 183 and Line under  “p=0.031.” What test does this probability refer to?

Line 216-224: “ How were the micro elements quantified? Are you sure there are all these insufficient concentrations or could there be a quantification bias?

Line 257 table 2: I would suggest to order the results in function of the “duration” (e.g. <2, 3-6, >7) for all the variables.

Moreover, why are the time frames different among the variables?

Line 257 table 2; Line 259 table 3 Line 266 table 4: statistics show too many chi square test, for which there is an excess of false positives. A log model is required. linear, instead of many chi square test.

Line 318 table 6: If there was no statistically significantly difference between the LBM of those taking and 315

not taking supplements (%) (Table 6), why F = 9.72 and p <0.0001? please, clarify!

Line 328 table 7: see line 316 table 6 (the same comment)

Discussion

Line 371: Reviewer suggest to compare data of the present study, with already reported data present in the literature. Amatori et al. 2020 (doi:10.3390/sports8030031);  - Heikkinen et al. 2011; - Heikura et al. 2017.

Given the above reported comments, reviewer suggests a major review in order to improve the quality of the paper, and maybe consider it for a possible publication in “Medicina”.

Author response to Reviewer 1 in new submission cover letter

Reviewer’s observation: Line 16: 76.7% of what? Total Lithuanian athletes?

Reply and corrections: "The research subjects were 76.7% of Lithuanian elite athletes (n=247)"

Reviewer’s observation: Line 33: Nutrition is partly determined by health, or health is partly determined by nutrition? Please, comment.

Reply and corrections: Health is partly determined by the state of one’s nutrition

Reviewer’s observation: What do authors mean with ‘only then’?

Reply and corrections: “An athlete’s diet should be coordinated with the training process, because then is it possible to improve physical working capacity during the pre-competition period and achieve better results in sports competitions”

Reviewer’s observation: Line 44: Not only the professional athletes need to reach the RDI. Maybe it would be the case to say that professional athletes, due to high workloads, have different RDI respect to “normal people”? Please, Comment

Reply and corrections: “Professional athletes, due to high workloads, therefore need to get enough energy and nutrients (carbohydrates, protein and fat) with food. Basically athletes do not have different recommended daily intake (RDI) for vitamins and minerals compared to the general population [5].”

Reviewer’s observation: Line 67: In which period of the season data have been collected? Body composition and dietary habits of the athletes may vary in function of the period (eg. off-season, preparatory, competitive).

Reply and corrections: Additionally indicated “The highly skilled athletes were tested during the preparatory period before a competition”

Reviewer’s observation: Furthermore, if data were collected between 2016 and 2017, it was a period following the Rio Olympics: could have this affected the results? Authors are suggested to discuss this point

Reply and corrections: “Plans specified in the Rio 2016, Sochi 2014 and Pyeongchang 2018 programmes.” Comment: As the data collection began in 2016, at that time the precise Pjongcang plan was not regulated, so, at the start of the research, the Sochi program was followed. Later, the data were adjusted and modified according to the Pjongcang plan. Essentially, the number of athletes in winter sports in Lithuania is small and does not change for at least 3 Olympic cycles, and the plans for training sessions of athletes for the Olympics are almost identical. Therefore, changes in the quantity and quality of the programs (Sochi and Pjongcang) did not affect the data collection and the course of the study.

Reviewer’s observation: Line 95: Was the weight of the muscle obtained by the BIA? Please, Report method used.

Reply and corrections. Additionally indicated: “Measurements of body weight and individual weight components (body mass, LBM (in kilograms and percentage), muscle mass (in kilograms and percentage) and body fat (BF) (in kilograms and percentage) were performed at the Lithuanian Sport Centre using the BIA (bioelectrical impedance analysis) tetra-polar electrodes and measuring resistivity with 8-12 tangent electrodes at different frequencies of signal: 5, 50 and 250 kHz.”

Reviewer’s observation: Line 106 Atlas foodstuffs: Please, report reference

Reply and corrections. Additionally, a reference is reported: [Barzda A, Bartkevičiūtė R, Viseckienė V, Abaravičius AJ, Stukas R. Maisto produktų ir patiekalų porcijų nuotraukų atlasas (Atlas of Foodstuffs and Dishes), Vilnius, Republican Nutrition Center. Vilnius University Faculty of Medicine. 2007. p. 7-42. http://www.smlpc.lt/media/file/Skyriu_info/Metodine_medziaga/Maisto%20prod%20atlasas%202007.pdf. Accessed 14 Apr 2020.]

Reviewer’s observation: Line 122: “…meets the prescribed RDI for a given country”. It is unclear. Please, clarify

Reply and corrections. Changed to “Athletes’ increased intake of vitamins and/or minerals has little or no effect on physical performance indicators as long as the dietary intake of vitamins and minerals meets the prescribed RDI for a given country. Thus, athletes should consume diets that provide at least the RDI for all micronutrients [5, 21, 23]. In the case of our study, the same RDI for vitamins and minerals (exclusively, iron and calcium RDI [5, 24, 25]) of the Lithuanian population [26] were also applied to the analysis of the diets of the studied athletes in Lithuania.”
Reference: [26] [2016 June 23 order No. V-836 of Minister of Health of the Republic of Lithuania. Recommended daily intake for energy and nutrients. Ministry of Health of the Republic of Lithuania. https://www.e-tar.lt/portal/lt/legalAct/4bd890f0428011e6a8ae9e1795984391. Accessed 14 Apr 2020.]

Reviewer’s observation: Line 145: Validated by…?

Reply and corrections. Changed to: We used validated (by Baranauskas M. [30]) questionnaire for the food supplementation study. Reference: [30] [Baranauskas M. Assessment of actual nutrition and dietary habits of athletes during the 2008-2012 Olympic period. Doctoral Dissertation: Faculty of Medicine of Vilnius University, Lithuania. p. 229-233. (The doctoral dissertation is available at the library of Vilnius University).]

Reviewer’s observation: Line 158: Consent to participate? In particular for <18 yo athletes

Reply and corrections. Additionally indicated: “Prior to testing, all athletes provided written consent and the study protocols were approved by the Lithuanian Sports Medicine Center Institutional Review Board.”

Reviewer’s observation: Line 165: Find a synonimum, please. Line 165: Why did you split into two age groups? by what criteria? Line 167: This has already been reported in the Methods section. Please remove one of the two, to avoid repetitions. Line 177-178: This is a discussion phrase. Please, move in discussion. Line 183 and Line under “p=0.031.” What test does this probability refer to? Line 216-224: “ How were the micro elements quantified? Are you sure there are all these insufficient concentrations or could there be a quantification bias?

Reply and corrections. All comments and suggestions for authors were improved.
Apart from that, we additionally calculated the mean±SD intake of vitamins and minerals by athletes and compared them with RDI. We used an additional paired samples t-test to compare the means to avoid errors.
In the results section, we marked where the chi-square test was used. Fisher’s exact test is used in unmarked places.

Reviewer’s observation: Line 257 table 2: I would suggest to order the results in function of the “duration” (e.g. <2, 3-6, >7) for all the variables. Moreover, why are the time frames different among the variables? Line 257 table 2; Line 259 table 3 Line 266 table 4: statistics show too many chi square test, for which there is an excess of false positives. A log model is required. linear, instead of many chi square test.

Reply and corrections. Due to the excess data quantity, we shortened the article, excluded the comparisons by age, and the analysis of the frequency of food supplement use during the competition. We did this because the use of dietary supplements in different subgroups by age and sports did not differ significantly.
In the absence of significant differences and consequently the absence of the relationship between the features under analysis, it is not recommended to analyse the data using the regression method. Also, using multivariate linear regression, the influence of different types of food supplements on LBM% and BF% has already been determined in the in the article (paragraph 3.3) “Interactions between dietary intake, food supplementation and body composition”.

Reviewer’s observation: Line 318 table 6: If there was no statistically significantly difference between the LBM of those taking and 315 not taking supplements (%) (Table 6), why F = 9.72 and p <0.0001? please, clarify! Line 328 table 7: see line 316 table 6 (the same comment)

Reply and corrections. Indicators of a good and appropriate regression model (R2 ≥ 0,20; ANOVA (F), P < 0,05) are defined according to “F = 9.72 and p <0.0001”. However, how the variables (LBM% and BF%) are impacted by the individual factors (e.g., protein content, energy value, or dietary supplements), are indicated by the coefficient of the beta (β) factor and the adjacent P value. Therefore, we presented data on the suitability of the regression model only below the tables.
Moreover, according to our data, LBM% are not affected by the use of different types of dietary supplements, (data adjusted by gender, branch of sports). This is explained by the fact that it is the diet of conventional foods that has a greater impact. In the specific case, consuming more protein with food (e.g. with meat, poultry, fish, dairy products, etc.) - about 2g/kg/daily, increases the probability of increasing the LBM% by 3.5%. Meanwhile, the effect is low with protein supplements (the most common single dose in practice is 0.25 g / kg).
The results of our study highlighted the much greater impact of normal diet on athletes’ body composition compared to the use of dietary supplements. Yet, in no way do we claim that dietary supplements are ineffective.

Reviewer’s observation: Line 371: Reviewer suggest to compare data of the present study, with already reported data present in the literature. Amatori et al. 2020 (doi:10.3390/sports8030031); - Heikkinen et al. 2011; - Heikura et al. 2017.

Reply and corrections. All comments and suggestions for authors about discussion were improved.

Author response to Reviewer 2 in new submission cover letter

Reviewer’s observation: A cross-sectional, observational study cannot be used to draw conclusions on cause and effect. Most importantly, the methods are severely limited and do not allow the authors to support their conclusions. Neither can the “effects of different nutrients on body composition” be ascertained from a correlative analysis, nor has sufficient thought been given to the limitations of 24 h dietary recall on the assessment of athletes’ intake. The limitations of the methods of dietary analyses and body composition assessment used prevent robust conclusions being made on these athletes’ diets, and have not even been mentioned, yet alone addressed. Insufficient evidence is provided to support the use of validated instruments (e.g. the neither the supplement questionnaire, nor portion-size resource are properly referenced), while the limitations to other methods (24 h dietary recall/bioelectrical impedance analysis) have not been discussed or overtly addressed.

Reply and corrections. We have carried out a study using multiple linear regression, the findings of which allow us to predict the impact of the use of individual macronutrients and dietary supplements in athletes’ LBM% (lean body mass) and BF% (body fat). In the cross-sectional, observational study it is possible and even recommended to perform linear multivariate regression in order to predict the effect of relevant factors and the features under analysis on the variables. Regression shows dependence but not relationship. We performed multiple linear regression, controlling data by gender, branch of sports, and age. The regression method used in our study was prognostically stronger than a simple correlation analysis (Pearson’s “r” or Spearman’s “rho”). In the article, we additionally indicated specific references to literature sources in which research methods used in other studies were confirmed (24-food recall, portion sizes, BIA, food habits questionnaire).

Reviewer’s observation: For example, when discussing the sub analysis on athlete protein intake, the authors switch between describing a “3.5% increase in LBM (%)” (line 323), to “3.5% of Lithuanian professional athletes” (line 433).From the data provided, it would seem they are referring to the regression analysis and inferring a 3.5 % greater amount of lean mass. However, the wording on line 323 sounds like they have tested at multiple time-points and observed an increase in LBM, whilst the wording on line 433 makes the 3.5 % seem like a subgroup of athletes

Reply and corrections. Multiple linear regression can predict a 3.5% higher LBM% when higher amounts of protein are consumed with food (excluding non-dietary supplements). The effect of the use of food supplements on the body composition of athletes has hardly been determined. Our method of analysis allowed us to determine the impact of nutrient intake on the entire athlete population without distinguishing individual subgroups. Because suitable (F (11, 224)=9.72, p<0.0001, R2=0.32; F(8, 238)=26.30, p<0.0001; R2=0.45; F(14, 221)=8.11, p<0.0001, R2=0.34) regression models were constructed by controlling the branch of sports, gender, age of athletes. In other words, the influence of the branch of sports, gender, and age on the assessment of the effects of nutrients and food supplements in LBM% (lean body mass) and BF% (body fat) was eliminated.

Reviewer’s observation: Furthermore, conclusions are made on athletes’ vitamin D status from only dietary analysis, which is not valid

Reply and corrections. We found a deficiency of vitamin D in the diet of athletes, which is described in our study. However, we did not find an active deficiency of vitamin D in the blood of the subjects. In the article, we supplemented the information about vitamin D as follows: “Vitamin D deficiency is common in Europe mostly countries in Northern European latitudes (>35° N) such as the UK, Ireland, Denmark, France, Germany, and Lithuania. Since Lithuanian high-performance athletes tend to consume little vitamin D from the diet and dietary interventions alone have not been shown to be a reliable means to resolve insufficient status, supplementation above the current RDI and/or responsible UVB exposure may be required to maintain sufficient vitamin D status. “

Reviewer’s observation: whilst European RNI values for n3/n6 fats are debateable anyway, the reference provided (ACSM 2016) doesn’t even provide such recommendations for PUFAs

Reply and corrections. Recommended by the literature source, p. 552: [Thomas DT, Erdman KA, Burke LM. American College of Sports Medicine Joint Position Statement. Nutrition and Athletic Performance. Med Sci Sports Exerc. 2016;48(3):543–68.]: “Athletes should be discouraged from chronic implementation of fat intakes below 20% of energy intake since the reduction in dietary variety often associated with such restrictions is likely to reduce the intake of a variety of nutrients such as fat-soluble vitamins and essential fatty acids, especially n-3 fatty acids. Intake of fat by athletes should be in accordance with public health guidelines and should be individualized based on training level and body composition goals“. We supplemented the list of references with RDI for SFA and PUFA. Therefore, our study followed the FAO / WHO recommendations: [FAO/WHO Expert Consultation on Fats and Fatty Acids in Human Nutrition. Fats and Fatty Acids in Human Nutrition: Report of an Expert Consultation; 10–14 November 2008. Geneva: Food and Agriculture Organization of the United Nations; 2010.]

Reviewer’s observation: The sentence “Incomplete restoration of glycogen storage in the muscles and liver between workouts requires more central nervous system effort to overcome high-intensity physical loads” is not supported with evidence (lines 40 – 41)

Reply and corrections. As indicated in the literature source mentioned in our article [Burke LM. Fuelling strategies to optimize performance: training high or training low? Scand J Med Sci Sports. 2010;20(2):48–58.] : 49 p. „An important finding from two of these investigations was that a higher daily carbohydrate intake was able to reduce, but not entirely prevent, the development of over-reaching symptoms (fatigue, impaired performance, sleep and mood disturbance, altered hormonal responses to stress, etc.), which can occur when a period of intensified training is undertaken.“ p. 54 (Table 3, rows 2-7,) “Chronic reduction in muscle carbohydrate availability (endogenous and potentially exogenous sources) for all training sessions, depending on degree of fuel mismatch. Chronic whole-body effects of low carbohydrate availability including impairment of immune system and central nervous system function.“

Reviewer’s observation: Numerous instances of incorrect/inappropriate English terminology further undermine the authors’ points; g. nutrition is “determined by the state of one’s health” (line 11), meaning that individuals/athletes will eat differently if they are more or less healthy? Why is this relevant? “blended (aerobic and anaerobic) force” (line 168) is incorrect, as it is power that is the application of force in a given time, and the word blended doesn’t work in this context (far too “physical”, suggesting these two components are tangible, physical objects that have been mashed and mixed.

Reply and corrections. Corrected the English terms as follows: “Health is partly determined by the state of one’s nutrition”, “Depending on the duration of the physical labour, training, and features of energy production in the body, we classified the athletes in accordance with the sport into three groups: anaerobic (21.9% (N=54)), mixed aerobic and anaerobic (30.8% (N=76)), and aerobic (47.4% (N=117))”

Reviewer’s observation: However, the research questions are not well defined and this has led to the inclusion of excessive amounts of data/comparisons that do not help answer clearly defined questions.

Reply and corrections. Due to the excessive amount of data, we shortened the article, abandoned the comparisons by age, and the peculiarities of the use of food supplements during the competition. In addition, we calculated the average intakes of vitamins and minerals and compared them with RDI. We used an additional paired samples t-test to compare the means. We also adapted the discussion section of the manuscript according to the adjusted results of the research.